# NUF2 Expression in Cancer Tissues and Lymph Nodes Suggests Post-Surgery Recurrence of Non-Small Cell Lung Cancer

**DOI:** 10.3390/diagnostics14050471

**Published:** 2024-02-21

**Authors:** Chika Shirakami, Koei Ikeda, Hironori Hinokuma, Wataru Nishi, Yusuke Shinchi, Eri Matsubara, Hironobu Osumi, Kosuke Fujino, Makoto Suzuki

**Affiliations:** Department of Thoracic Surgery, Graduate School of Medical Sciences, Kumamoto University, Kumamoto 860-8556, Japan; sachika.x.gratitude2@gmail.com (C.S.); hironori.hinokuma@gmail.com (H.H.); kashiragata@gmail.com (Y.S.); blueberry04090302@gmail.com (E.M.); h-osumi@kumamoto-u.ac.jp (H.O.); kfujino@kumamoto-u.ac.jp (K.F.); smakoto@kumamoto-u.ac.jp (M.S.)

**Keywords:** NUF2, lung cancer, lymph node metastasis, sublobar resection, intraoperative, non-small cell lung cancer

## Abstract

In non-small cell lung cancer (NSCLC) cases, detecting potential lymph node metastases is essential to determine the indications for sublobar resection or adjuvant therapy. NUF2 is a tumor-specific antigen that is highly expressed in lung cancer tissues. However, the significance of analyzing NUF2 expression in dissected lymph nodes has not yet been studied. Thus, we investigated the association between NUF2 expression in lung cancer tissues and dissected lymph nodes and early recurrence of NSCLC to determine its usefulness as a marker of lymph node micrometastasis. This retrospective study quantified NUF2 expression in the cancer tissues of 88 patients with NSCLC who underwent complete resection using real-time polymerase chain reaction and investigated its relationship with clinicopathological features and prognosis. We also quantified NUF2 RNA expression in mediastinal lymph nodes from 255 patients with pN0 NSCLC who underwent complete resection with lymph node dissection and analyzed its association with prognosis. NUF2 expression in primary tumors was correlated with lymph node metastasis and unfavorable outcomes in terms of poor recurrence-free and cancer-specific survival. In N0 NSCLC cases, high NUF2 expression in mediastinal lymph nodes indicated poor prognosis, especially in lymph node recurrence. NUF2 emerges as a promising marker for predicting lymph node metastatic recurrence, offering potential utility in guiding post-surgical adjuvant therapy for lung cancer or assisting in intraoperative decisions for sublobar resection.

## 1. Introduction

Among all cancers in Japan, lung cancer is the most common cause of death. Stages IA3 and IB have 5-year recurrence-free survival rates of 74.8 and 71.3%, respectively [1], and even N0 lung cancer recurs to a certain extent. Bearing significant prognostic implications, lymph node recurrence underscores the need to identify potential lymph node metastases to determine the indication for adjuvant therapy or sublobar resection. Intraoperative rapid pathological diagnosis is the primary method used in practice. However, since the diagnosis is made mainly on a single slice, metastases may be overlooked due to the localization of metastatic lesions. For diagnosing rapid intraoperative lymph node metastasis, the one-step nucleic acid amplification (OSNA) method is widely used for skin and breast cancers. In Japan, the OSNA method for lung cancer has recently been covered by insurance [2,3]. However, the OSNA method targets the epithelial marker CK19, which may lead to false positive results, especially in the rapid diagnosis of hilar lymph nodes. Therefore, exploring new molecular markers to detect lung cancer metastases is necessary.

NUF2, previously known as cell division cycle-associated protein 1 (CDCA1), is a tumor-associated antigen of lung cancer that was identified from a genome-wide cDNA array by Nakamura et al. in 2008 [4]. This molecule is expressed in various malignancies; however, other than in the testis, it is not expressed in normal adult tissues, including the lung tissues. Thus, NUF2 is recognized as a type of cancer–testis antigen. NUF2 is a component of the NDC80 complex and is involved in microtubule stabilization in the kinetochore [5,6]. NUF2 is a critical cell cycle regulator and has been reported to be a prognostic factor for several cancers [7,8,9,10,11].

Several papers have reported an association between NUF2 expression and prognosis in NSCLC. However, no report has examined the significance of analyzing NUF2 expression in dissected lymph nodes. This study investigated the relationship between NUF2 expression in the cancer tissues of patients with NSCLC who underwent curative-intent resection and their clinicopathological factors and prognosis. In addition, we investigated the relationship between NUF2 expression in the dissected lymph nodes of patients with pathologic N0 NSCLC and their prognosis to determine its usefulness as a lymph node micrometastasis marker.

## 2. Materials and Methods

### 2.1. Patients and Samples

This study was approved by the Ethics Committee of the Graduate School of Life Sciences at Kumamoto University (ID Ethics No. 2035). Written informed consent was obtained from the participants while undergoing surgery at the Department of Thoracic Surgery, Kumamoto University Hospital. Normal lung tissues and lung cancer tissues were collected from 88 patients who underwent complete resection for NSCLC from 1 January to 31 December 2014 at Kumamoto University Hospital (tumor cohort). Patients with a history of treatment for other cancers and who received neoadjuvant chemotherapy or radiation therapy were excluded from the analysis. Most patients were older adults and had a history of non-cancer diseases. All patients were staged with preoperative positron emission tomography–computed tomography and brain magnetic resonance imaging. No patient had a preoperative or intraoperative pathologic diagnosis of lymph node metastasis. The patient characteristics are summarized in Table 1. A small amount (5 mm^3^) of lung cancer tissue was taken from the lesion of the explanted specimen, avoiding the largest diameter to not affect the pathological diagnosis. Tumor sampling was performed in the center of the tumor as much as possible, to avoid affecting the histopathology diagnosis. Normal lung tissues were collected from the area farthest from the lesion in the explanted specimen. Lymph nodes were also collected from 255 patients who underwent complete resection with lymph node dissection for pN0 NSCLC between 1 October 2017 and 31 September 2020 (lymph node cohort). Forty-three patients underwent complete mediastinal dissection and 212 underwent selective lymph node dissection. The mean number of lymph nodes dissected was 13.4 (2–44). The number of hilar lymph nodes was 6.0 (1–21) and mediastinal lymph nodes was 7.1 (1–41). One of the dissected mediastinal lymph nodes was selected and partially cryopreserved, and the majority were subjected to histopathological analysis. The choice of lymph nodes depended on the extent of selective dissection for the lobe of the tumor, that is, right lower paratracheal nodes (#4R) for the right upper lobe, para-aortic nodes (#5) for the left upper lobe, sub-carinal nodes (#7) for the lower lobe, and right lower paratracheal nodes (#4R) or sub-carinal nodes (#7) for the middle lobe. The largest lymph nodes of the station were selected. The samples were placed in RNA later^®^ (QIAGEN, Hilden, Germany) and frozen at −80 °C for stabilization. The patient characteristics are summarized in Table 2.

### 2.2. Cell Line

H358 was purchased from ATCC (Manassas, VA, USA) and was grown in RPMI 1640 medium supplemented with 10% fetal bovine serum. H358 cells were incubated at 37 °C in 5% CO_2_ and saturated humidity.

### 2.3. Reverse Transcription Polymerase Chain Reaction

Total RNA was extracted from the cryopreserved tissues (normal lung tissues, lung cancer tissues, and lymph nodes) collected from the resected specimens using the Total RNA Extraction Miniprep System ver.17A (Viogene, New Taipei City, Taiwan), according to the manufacturer’s protocol. Complementary DNA (cDNA) was generated from the extracted total RNA. RNA extracted from the explanted specimens was reverse-transcribed using ReverTra Ace^®^ qPCR RT Master Mix (Toyobo, Osaka, Japan). Quantitative (real-time) reverse transcription polymerase chain reaction (RT-PCR) was performed using Applied Biosystems TaqMan Gene expression assays (Thermo Fisher Scientific, Tokyo, Japan; NUF2, Hs00230097_m1 and GAPDH, Hs02758991_g1). Samples and reagents were loaded into each well as follows: 1 μL cDNA sample, 0.5 μL 20× Gene Expression Assay, 5 μL Gene Expression Master mix (Applied Biosystems, Vilnius, Lithuania), and 3.5 μL nuclease-free water. Reactions were performed in duplicate in 384-well plates using a QuantStudio 12 K Flex system (Thermo Fisher Scientific, Waltham, MA, USA). The PCR reaction conditions were as follows: first step, 50 °C 2 min; second step, 95 °C 10 min; third step, 95 °C 15 s; and 60 °C 1 min repeated for 40 cycles. Relative expression (RE) was calculated using GAPDH as the intrinsic control and RNA extracted from the H358 lung adenocarcinoma cell line as the calibration sample. The calculation formula is as follows: Deltadelta Ct = (Ct (NUF2 of sample) − Ct (GAPDH of sample)) − (Ct (NUF2 of H358mRNA) − Ct (GAPDH of H358mRNA)), Relative expression (RE) = 2-deltadelta Ct.

### 2.4. Statistical Analysis

The data are expressed as mean ± standard deviation for continuous variables and numbers and percentages for categorical variables. All statistical analyses were performed using SPSS for Windows (version 15; Texas Instruments, Dallas, TX, USA). Differences between continuous variables were evaluated using unpaired 2-tailed Student’s *t*-tests, and categorical data were compared using chi-square tests. NUF2 expression levels between lung cancer tissues and normal lung tissues were compared using the Wilcoxon signed-rank test. Odds ratios, 95% confidence intervals, and corresponding *p*-values were analyzed. Statistical significance was set at *p* < 0.05. The Kaplan–Meier method and log-rank test were used to assess the survival time distribution for survival analysis. A multivariate, stage-stratified Cox proportional hazard model was constructed to compute the hazard ratio based on the expression levels of NUF2, including sex (man vs. woman), age at surgery (≥75 vs. <75 years), tobacco use (yes vs. no), histological grade (G1–2 vs. G3), and pathological stage (Stage I vs. Stages II and III). The interaction was assessed by including the cross-product of the NUF2 variable and another variable of interest in a multivariate Cox model; thereafter, the Wald test was performed.

## 3. Results

### 3.1. NUF2 Expression in NSCLC Tissues

We performed quantitative RT-PCR analysis of NUF2 gene expression in cancer and corresponding normal lung tissues isolated from 88 patients with NSCLC. The results indicated that the NUF2 expression level was significantly higher in the NSCLC tissues than in the normal lung tissues (Figure 1, 0.21 ± 0.024 vs. 0.083 ± 0.007, *p* < 0.001). 

Table 3 shows the relationship between NUF2 expression levels and clinicopathological factors. NUF2 expression was higher in patients with lymph node metastasis (*p* = 0.034). There was no significant relationship between the expression levels and age, sex, histology, tumor size, and pathological grade. Figure 2 shows the correlation between NUF2 expression in tumor tissues and patient survival. We set the threshold to 0.210 between NUF2 high- (24 cases) and low-expression cases (64 cases) according to the mean RE. Patients with lung cancer whose tumors showed higher NUF2 expression had shorter survival than those with lower NUF2 expression both in recurrence-free survival (*p* = 0.03, by the log-rank test; Figure 2A) and cancer-specific survival (*p* < 0.001; Figure 2B). Five-year recurrence-free and cancer-specific survival rates were 79.0% and 90.8% in NUF2-low cases and 56.3% and 58.3% in NUF2-high cases, respectively. We also applied univariate analysis to evaluate associations between patient recurrence and other factors, including sex (male), age (≥75 years), smoking status (positive), histologic type (non-adenocarcinoma), lymph node metastasis (positive), solid tumor size (>3 cm), pathological grade (≥2), pleural invasion (plus), vascular invasion [v(+) or ly(+)], and NUF2 status (>0.210). Among these variables, NUF2 status (*p* = 0.004) and sex, histology, lymph node metastasis, pathological grade, and pleural and vascular invasion were significantly associated with poorer prognosis (Table 3). In the multivariate analysis between recurrence-free survival and the prognostic factors, NUF2 expression (*p* = 0.008) and sex, histology, lymph node metastasis, and pleural invasion were identified as independent prognostic factors (Table 4). Among 65 adenocarcinoma cases, the proportion of adenocarcinomas with a micropapillary component was 9 out of 27 (33.3%) in cases with high NUF2 expression, which was significantly higher than 1 out of 38 (0.03%) in cases with low NUF2 expression (*p* < 0.001).

### 3.2. NUF2 Expression in Resected Mediastinal Lymph Nodes of Patients with pN0 NSCLC

The NUF2 expression in dissected lymph nodes of patients with pN0 NSCLC was measured. The association between NUF2 expression levels and clinicopathological factors of patients is shown in Table 5. The NUF2 expression level was higher in patients aged > 75 years than in the others. The other factors showed no relationship with NUF2 expression. Figure 3 shows the correlation between NUF2 expression in dissected lymph nodes and patient prognosis. We set the threshold as 0.128 between NUF2 high- (69 cases) and low-expression cases (186 cases) according to the receiver operating characteristic curve for recurrence. Patients with lung cancer whose tumors showed higher NUF2 expression had shorter survival than those with lower NUF2 expression in recurrence-free survival (*p* = 0.03, by the log-rank test; Figure 3A) and cancer-specific survival (*p* = 0.01, Figure 3B). The 5-year recurrence-free and cancer-specific survival rates were 76.5% and 96.1% in NUF2-low cases and 76.9% and 85.0% in NUF2-high cases, respectively. In the multivariate analysis between recurrence-free survival and the prognostic factors, NUF2 expression (*p* = 0.03) and sex (male), histology (non-adenocarcinoma), pleural invasion, and vascular invasion were identified as independent prognostic factors (Table 6). Among the recurrent cases, 9 of 14 cases (64.3%) had mediastinal lymph node metastasis at the time of first recurrence in the NUF2 high-expression cases. On the other hand, mediastinal lymph node recurrence was identified in only 4 of 16 cases (25.0%) in the NUF2 low-expression cases (*p* = 0.03, Figure 4).

## 4. Discussion

NUF2 is a member of the NUF2 complex, a protein involved in stabilizing the kinetochore during cell division [5,6]. Its overexpression has been reported across various cancers, including colorectal, prostate, breast, and gastric cancers and melanoma, where it serves as a prognostic indicator of poor outcomes [7,8,11,12,13]. Similar results have been reported for NSCLC [9,10]. Inhibition of NUF2 expression was reported to suppress the growth of cancer cells and induce apoptosis [4,14]. In this study, NUF2 was highly expressed in NSCLC tissues, correlating significantly with lymph node metastasis and indicating a significant association with poor prognostic factors. Furthermore, we showed that high NUF2 expression in dissected lymph nodes was a significant poor prognostic factor in N0 NSCLC. To our knowledge, there are no publications showing that NUF2 expression in dissected lymph nodes is associated with prognosis in NSCLC.

Many types of molecular techniques have been developed to detect microscopic cancer cells. Typical examples include PCR, immunohistochemistry, flow cytometry, immunomagnetic beads, and OSNA [15,16,17,18,19,20]. Recently, methods have also been developed using nanoenzymes [21], liquid biopsies targeting cell-free nucleic acids [22], and long non-coding RNAs [23]. Among those used for intraoperative lymph node metastasis diagnosis, one of the most representative methods is the measurement of carcinoembryonic antigen mRNA levels in lymph nodes; however, this method has not been put into practical use until now [24,25]. Currently, OSNA is the most generalized intraoperative lymph node metastasis detection method. Its effectiveness has also been reported in lung cancer [2,3]. OSNA takes approximately 40 min from lymph node processing to reporting, and OSNA has been reported to save USD 346 per patient in intraoperative sentinel lymph node biopsies for breast cancer in Japan [26]. Confirming metastasis in the hilar lymph nodes is necessary to indicate sublobar resection for small-sized NSCLC. However, CK19, the target gene of the OSNA assay, may give false-positive results because it is also expressed in normal lung tissue. Even with careful sampling, normal lung tissue can be contaminated when collecting hilar lymph nodes. NUF2 is a good candidate for a new marker for detecting lymph node metastasis because it is frequently expressed in lung cancer tissue and not in lung and other normal tissues [9]. In fact, we assessed CK19 expression in the same sample of the lymph node cohort and found no correlation with prognosis. If we can develop OSNA with NUF2, it can probably be operated at the same time and cost as CK19. 

To our knowledge, this is the first study to show that high NUF2 expression in mediastinal lymph nodes is a significant early recurrence marker in NSCLC in cases of N0 radical resection. In a review of postoperative recurrence cases, the majority of the recurrences occurred in the mediastinal lymph nodes. Considering that the frequency of NUF2 expression in resected lymph nodes is significantly related to recurrence rates, it is possible that micrometastases that cannot be detected by conventional pathology may be detected. Lymph node recurrence was the most common type in patients with high NUF2 levels. These results suggest high NUF2 expression in mediastinal lymph nodes may detect micrometastases missed by histopathological diagnosis. Few articles have described risk factors for lymph node recurrence in NSCLC. Vascular invasion within the tumor and high expression of CUB domain-containing protein (CDCP1) and low levels of insulin-like growth factor binding protein (IGFBP5 and IGFBP7) in the bloodstream have been reported as risk factors for lymph node recurrence [27,28,29]. Analysis of NUF2 expression in dissected lymph nodes may predict postoperative lymph node recurrence in early-stage lung cancer and determine the indication for adjuvant chemotherapy after surgery. 

This study had several limitations. First, it was a retrospective study of surgical cases at a single institution. In addition, the lymph nodes analyzed by PCR were a subset of the lymph nodes, not the entire lymph node. In the OSNA method, the whole lymph node is processed to determine the expression of CK19; however, in this study, a portion of the dissected lymph node was cryopreserved, and the maximum allocation of the specimen was made to histological metastasis determination, which may have induced a false negative result. In future studies, we aim to consider a prospective study in which the entire lymph node is processed and NUF2 expression is determined intraoperatively simultaneously with the OSNA method. In this study, the tumor area was not evaluated by immunostaining. This is because the correlation between immunostaining results and prognosis has already been reported [4].

## 5. Conclusions

High NUF2 expression in tumor tissue has a negative impact on tumor prognosis. Additionally, high NUF2 expression in the mediastinal lymph nodes of patients with N0 NSCLC is a marker of early tumor recurrence. This suggests the possibility of detecting tumor micrometastasis by quantifying NUF2 expression in lymph nodes, which could be applied to intraoperative rapid diagnosis of lymph node metastasis while deciding on the indication for sublobar resection. Further analysis of the results of this study is warranted.

## Figures and Tables

**Figure 1 diagnostics-14-00471-f001:**
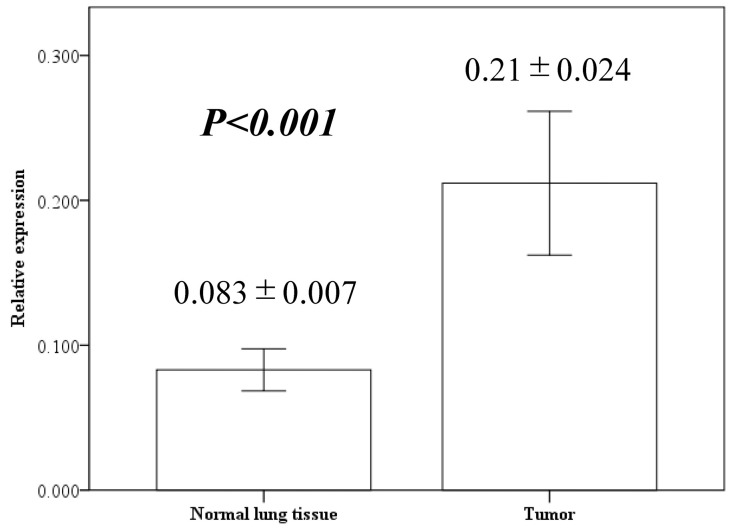
NUF2 expression in resected lung cancer tissues and adjacent normal lung tissues using quantitative reverse transcription polymerase chain reaction. Bar height indicates average value; whiskers indicate standard error.

**Figure 2 diagnostics-14-00471-f002:**
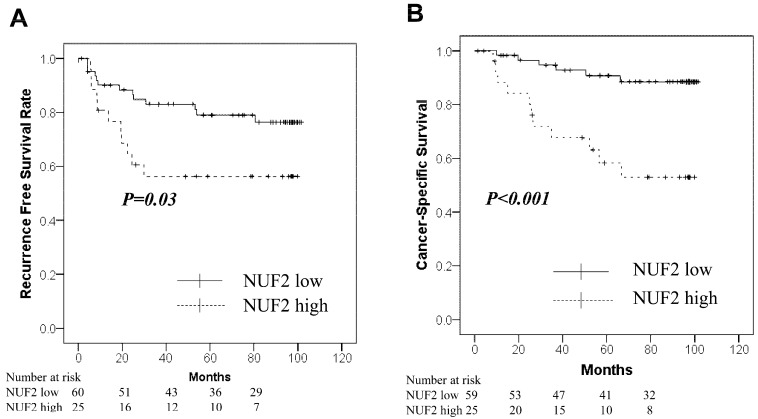
Kaplan–Meier curves for the NUF2-low group (solid line) vs. the NUF2-high group (dashed line) in the tumor cohort. (**A**): Recurrence-free survival. (**B**): Cancer-Specific survival.

**Figure 3 diagnostics-14-00471-f003:**
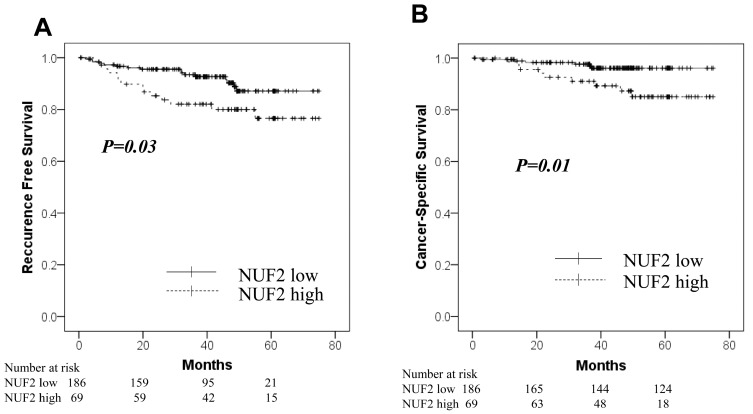
Kaplan–Meier curves for the NUF2-low group (solid line) vs. the NUF2-high group (dashed line) in the lymph node cohort. (**A**): Recurrence-free survival. (**B**): Cancer-Specific survival.

**Figure 4 diagnostics-14-00471-f004:**
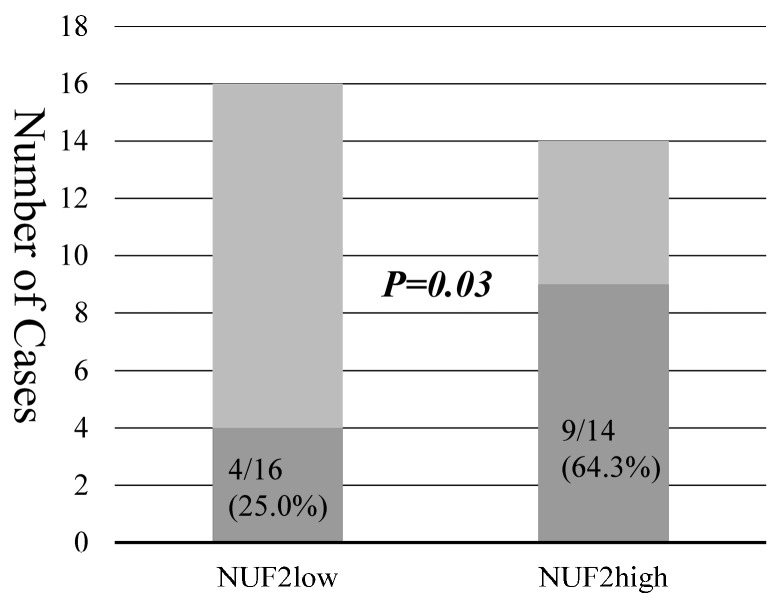
The bars show the numbers of recurrent cases and mediastinal lymph node metastasis in NUF2 low- and high-expression cases. Darker bars represent the number of recurrent cases with lymph node metastasis.

**Table 1 diagnostics-14-00471-t001:** Patient characteristics of the tumor cohort.

Characteristic	Patients (*n* = 88)
Age, y	70 (39–86)
Gender	
Male	45 (51.1)
Female	43 (48.9)
Smoking status (BI ≥ 200)	
Negative	39 (44.3)
Positive	49 (55.7)
Histology	
Adenocarcinoma	65 (73.9)
Squamous cell carcinoma	17 (19.3)
Adenosquamous carcinoma	4 (4.5)
Mucoepidermoid carcinoma	2 (2.3)
Lymph node metastasis	
pN0	71 (80.7)
pN1	7 (8.0)
pN2	10 (11.4)
Solid size	
≤3 cm	58 (65.9)
>3 cm	30 (34.1)
Pleural invasion	
Negative	67 (76.1)
Positive	21 (23.9)
pStage (8)	
IA1, 1A2	44 (50.0)
IA3, IB	20 (22.7)
IIA, IIB	11 (12.5)
IIIA, IIIB	13 (14.8)
Vascular invasion (ly or v)	
Negative	68 (77.3)
Positive	20 (22.7)
Pathological grade	
G1	20 (22.7)
G2	48 (54.5)
G3	20 (22.7)
Surgical procedure	
Pneumonectomy	3 (3.4)
Lobectomy	76 (86.4)
Bilobectomy	1 (1.1)
Segmentectomy	6 (6.8)
Wedge resection	2 (2.2)

Data are presented as median (range) or *n* (%).

**Table 2 diagnostics-14-00471-t002:** Patient characteristics of the lymph node cohort.

Characteristic	Patients (*n* = 255)
Age, y	71 (42–90)
Gender	
Male	139 (54.5)
Female	116 (45.5)
Smoking status (BI ≥ 200)	
Negative	108 (42.4)
Positive	147 (57.6)
Histology	
Adenocarcinoma	201 (78.8)
Squamous cell carcinoma	48 (18.8)
Adenosquamous carcinoma	5 (2.0)
Large cell carcinoma	1 (0.4)
Solid size	
≤3 cm	220 (86.3)
>3 cm	35 (13.7)
Pleural invasion	
Negative	221 (86.7)
Positive	34(13.3)
pStage (8)	
IA1	47 (18.4)
IA2	108 (42.4)
IA3	32 (12.5)
IB	56 (22.0)
IIA	4 (1.6)
IIB	8 (3.1)
Vascular invasion (ly or v)	
Negative	214 (83.9)
Positive	41 (16.1)
Pathological grade	
G1	37 (14.5)
G2	193 (75.7)
G3	24 (9.4)
G4	1 (0.4)
Surgical procedure	
Lobectomy	235 (92.2)
Bilobectomy	2 (0.8)
Segmentectomy	18 (7.1)

Data are presented as median (range) or *n* (%).

**Table 3 diagnostics-14-00471-t003:** Association between NUF2 expression in cancer tissue and patient characteristics in the tumor cohort (*n* = 88).

Characteristic		Number	NUF2 RE	*p*-Value
Age	≤75	64	0.20 ± 0.21	
	>75	24	0.23 ± 0.29	0.534
Gender	Male	45	0.24 ± 0.24	
	Female	43	0.18 ± 0.22	0.215
Smoke	Positive	49	0.24 ± 0.23	
	Negative	39	0.17 ± 0.23	0.132
Histology	Adenocarcinoma	65	0.20 ± 0.26	
	Non-Adenocarcinoma	23	0.25 ± 0.14	0.314
Lymph node metastasis	Metastasis (−)	71	0.19 ± 0.20	
	Metastasis (+)	17	0.32 ± 0.32	0.034
Tumor size (solid)	≤3 cm	58	0.22 ± 0.27	
	>3 cm	30	0.20 ± 0.13	0.677
Pleural invasion	Negative	67	0.20 ± 0.20	
	Positive	21	0.26 ± 0.32	0.276
Vascular invasion	Negative	68	0.21 ± 0.26	
	Positive	20	0.23 ± 0.11	0.624
Grade	≤2	68	0.20 ± 0.25	
	3	20	0.27 ± 0.13	0.241

RE, relative expression.

**Table 4 diagnostics-14-00471-t004:** Cox’s proportional hazards model analysis of prognostic factors in the tumor cohort.

	Univariate			Multivariate		
	HR	95% CI	*p*	HR	95% CI	*p*
Age (≥75)	1.05	0.42–2.65	0.92			
Sex (male)	4.65	1.73–12.48	0.003	3.71	1.20–11.46	0.02
Smoking (positive)	2.17	0.90–5.24	0.08			
Histology (non-AD)	2.65	1.17–5.98	0.02	0.16	0.42–0.65	0.01
Lymph node metastasis	7.17	3.18–16.15	<0.001	13.12	3.43–50.16	<0.001
Solid size (>3 cm)	1.69	0.75–3.81	0.21			
Pathological grade (≥2)	3.34	1.50–7.48	0.03	1.99	0.64–6.18	0.23
Pl (+)	5.30	2.36–11.90	<0.001	6.84	2.31–20.29	0.001
Vascular invasion (+)	3.28	1.45–7.40	0.04	0.33	0.81–1.36	0.12
NUF2 (high)	3.65	1.51–8.84	0.004	3.80	1.46–9.90	0.006

**Table 5 diagnostics-14-00471-t005:** Association between NUF2 expression in mediastinal lymph nodes and patient characteristics in the lymph node cohort (*n* = 255).

Characteristic		Number	NUF2 RE	*p*-Value
Age	≤75	172	0.10 ± 0.11	
	>75	83	0.08 ± 0.08	0.03
Gender	Male	139	0.89 ± 0.10	
	Female	116	0.10 ± 0.09	0.38
Smoke	Positive	147	0.09 ± 0.10	
	Negative	108	0.09 ± 0.09	0.90
Histology	Adenocarcinoma	201	0.09 ± 0.09	
	Non-Adenocarcinoma	54	0.11 ± 0.12	0.22
Tumor size (solid)	≤3 cm	220	0.09 ± 0.10	
	>3 cm	36	0.11 ± 0.09	0.28
Pleural invasion	Negative	221	0.09 ± 0.10	
	Positive	34	0.09 ± 0.09	0.77
Vascular invasion	Negative	214	0.09 ± 0.10	
	Positive	41	0.09 ± 0.09	0.83
Grade	≤2	230	0.09 ± 0.09	
	>2	25	0.10 ± 0.12	0.83

RE, relative expression.

**Table 6 diagnostics-14-00471-t006:** Cox’s proportional hazards model analysis of prognostic factors in the lymph node cohort.

	Univariate			Multivariate		
	HR	95% CI	*p*	HR	95% CI	*p*
Age (≥75)	1.35	0.64–2.83	0.43			
Sex (male)	3.86	1.57–9.45	0.003	3.21	1.04–9.90	0.04
Smoking (positive)	3.25	1.33–7.96	0.01	0.93	0.29–2.94	0.90
Histology (non-AD)	3.84	1.87–7.87	<0.001	2.36	1.06–5.24	0.04
Solid size (>3 cm)	2.58	1.15–5.79	0.02	1.50	0.63–3.53	0.36
Pathological grade (≥2)	0.65	0.16–2.73	0.56			
Pl (+)	4.47	2.09–9.58	<0.001	3.19	1.42–7.18	0.005
Vascular invasion (+)	4.46	2.11–9.42	<0.001	2.77	1.23–6.23	0.01
NUF2 (high)	2.17	1.06–4.47	0.03	2.30	1.06–4.97	0.03

## Data Availability

Data are contained within the article.

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
