# Peer review of "NUF2 Expression in Cancer Tissues and Lymph Nodes Suggests Post-Surgery Recurrence of Non-Small Cell Lung Cancer"

_diagnostics, 2024, doi:10.3390/diagnostics14050471_

Round 1

Reviewer 1 Report

Comments and Suggestions for Authors

While the study's findings present NUF2 expression as a promising prognostic marker for predicting lymph node metastatic recurrence in post-surgery non-small cell lung cancer (NSCLC), there are several areas that warrant further exploration.

Firstly, leveraging publicly available databases to explore the association between NUF2 expression and histopathological grading in lung cancer can provide broader validation within larger patient cohorts. Additionally, incorporating single-cell RNA sequencing data to identify specific cell types exhibiting high NUF2 expression will elucidate its role at a more granular level in tumor progression and metastasis.

Supplementing the study with protein-level experiments, such as immunohistochemistry or immunofluorescence, would strengthen the conclusion by providing visual evidence of NUF2 expression in primary tumor samples. Moreover, detailed images displaying NUF2 cellular localization within primary tumor specimens could offer insights into its potential role in promoting metastasis.

Furthermore, proposing avenues for future research, including the exploration of molecular mechanisms underlying NUF2-mediated metastasis and its potential as a therapeutic target, would further enhance the significance and impact of the study.

Author Response

Firstly, leveraging publicly available databases to explore the association between NUF2 expression and histopathological grading in lung cancer can provide broader validation within larger patient cohorts. Additionally, incorporating single-cell RNA sequencing data to identify specific cell types exhibiting high NUF2 expression will elucidate its role at a more granular level in tumor progression and metastasis.

Thank you for your valuable suggestion. We will certainly consider conducting analyses using publicly available databases and single-cell RNA sequencing data in future studies.

Supplementing the study with protein-level experiments, such as immunohistochemistry or immunofluorescence, would strengthen the conclusion by providing visual evidence of NUF2 expression in primary tumor samples. Moreover, detailed images displaying NUF2 cellular localization within primary tumor specimens could offer insights into its potential role in promoting metastasis.

Thank you for these excellent recommendations. We did, in fact, perform immunohistochemical analyses on our lung cancer tissue samples, and positive results were found in most of the specimens. Since a quantitative evaluation was difficult, we performed our evaluation via RNA expression levels in this study.

Furthermore, proposing avenues for future research, including the exploration of molecular mechanisms underlying NUF2-mediated metastasis and its potential as a therapeutic target, would further enhance the significance and impact of the study.

Thank you for this astute observation. This function of NUF2 has already been analyzed in other studies in this field, which we did not focus on in our study. However, its role as a marker for detecting micrometastases in lymph nodes was, indeed, examined in our study.

Reviewer 2 Report

Comments and Suggestions for Authors

1. Originality: Does the case contain new and significant information adequate to justify publication?

This study focuses on investigating the potential utility of NUF2 expression with real-time PCR as a marker for lymph node micrometastasis and its association with early recurrence of NSCLC cases.

While there are previous studies demonstrating NUF2 expression as a poor prognostic factor for NSCLC, this study appears to focus specifically on evaluating NUF2 expression in both primary tumor tissues and dissected lymph nodes and its association with early recurrence and prognosis. This approach could contribute novel insights into the utility of NUF2 as a biomarker. 

2. Relationship to Literature: Does the paper demonstrate an adequate understanding of the relevant literature in the field and cite an appropriate range of literature sources? Is any significant work ignored?

In general, it seems to refer to large and important works from the literature.

3. Methodology: Is the paper's argument built on an appropriate base of theory, concepts or other ideas? Has the research or equivalent intellectual work on which the paper is based been well designed? Are the methods employed appropriate?

This is a retrospective study analyzing NUF2 expression in NSCLC tissues and dissected lymph nodes and investigating its association with clinicopathological features and patient prognosis. Methodology, patients' characteristics and previous treatment histories are explained clearly. 

However, in the study, the superiority of the NUF2 method over CK19 detection using OSNA has been suggested. This study has only shown that patients with negative lymph nodes who expressed high NUF2 are more prone to recurrence. To further elaborate, it is essential to conduct a more comprehensive analysis showing the performance of NUF2 in identifying positive lymph nodes, too. Such analysis would provide better insights into the sensitivity and specificity of this method, which are critical for determining their clinical utility and comparing these methods. Would the authors explain why they didn’t consider including that analysis. Also, the authors could specify why different threshold determination methods were used for lymph node and tumor samples.

4. Results: Are results presented clearly and analyzed appropriately? Do the conclusions adequately tie together the other elements of the paper?

I commend the authors for their comprehensive and richly informative manuscript, particularly for their extensive use of tables and graphs. It would be more appropriate to maintain a consistent style for all tables. Therefore, I suggest that Table 6 be reorganized accordingly. Additionally, synchronizing the "number at risk" data in the Kaplan-Meier graphs with the timepoints would enhance clarity.

In patients with negative lymph nodes, the data on recurrence risk based on NUF2 expression show promising potential for predicting especially lymph node recurrence. However, the sensitivity of this marker in detecting recurrence is approximately 45%, while the specificity for detecting lymph node recurrence is around 70%. Whether this marker alone is sufficient for effective decision-making regarding adjuvant therapy needs to be reassessed from this perspective.

Analyzing the association between NUF2 expression and tumor size and lymph node positivity provided valuable insights from the study data highlighting the importance of NUF2 expression regardless of these factors. Adding a comparison with stage could further enhance the understanding if these data are available.

5. Implications for research, practice and/or society: Does the paper identify clearly any implications for research, practice and/or society? Does the paper bridge the gap between theory and practice? How can the research be used in practice (economic and commercial impact), in teaching, to influence public policy, in research (contributing to the body of knowledge)? What is the impact upon society (influencing public attitudes, affecting quality of life)? Are these implications consistent with the findings and conclusions of the paper?

The implications and phases of the conducted framework provide promising insights. We still need further prospective controlled studies to use this data in clinical practice. It is recommended to provide data on circulating tumor DNA (ctDNA), which is one of the rapidly integrating markers into current clinical practice.

6. Quality of Communication: Does the paper clearly express its case, measured against the technical language of the fields and the expected knowledge of the journal's readership? Has attention been paid to the clarity of expression and readability, such as sentence structure, jargon use, acronyms, etc.

The language used in the article is fluent and understandable, and apart from a few typos, there do not seem to be any major problems.

-Line 145. Figure 1. NUF2 expression in resected lung cancer tissues and adjacent normal lung tissues using quantitative reverse transcription polymerase chain reaction. Bar height indicates average value; whiskers ‘indicatestandard’ error.

Author Response

  1. Originality: Does the case contain new and significant information adequate to justify publication?

This study focuses on investigating the potential utility of NUF2 expression with real-time PCR as a marker for lymph node micrometastasis and its association with early recurrence of NSCLC cases.

While there are previous studies demonstrating NUF2 expression as a poor prognostic factor for NSCLC, this study appears to focus specifically on evaluating NUF2 expression in both primary tumor tissues and dissected lymph nodes and its association with early recurrence and prognosis. This approach could contribute novel insights into the utility of NUF2 as a biomarker.

  1. Relationship to Literature: Does the paper demonstrate an adequate understanding of the relevant literature in the field and cite an appropriate range of literature sources? Is any significant work ignored?

 In general, it seems to refer to large and important works from the literature.

  1. Methodology: Is the paper's argument built on an appropriate base of theory, concepts or other ideas? Has the research or equivalent intellectual work on which the paper is based been well designed? Are the methods employed appropriate?

This is a retrospective study analyzing NUF2 expression in NSCLC tissues and dissected lymph nodes and investigating its association with clinicopathological features and patient prognosis. Methodology, patients' characteristics and previous treatment histories are explained clearly.

However, in the study, the superiority of the NUF2 method over CK19 detection using OSNA has been suggested. This study has only shown that patients with negative lymph nodes who expressed high NUF2 are more prone to recurrence. To further elaborate, it is essential to conduct a more comprehensive analysis showing the performance of NUF2 in identifying positive lymph nodes, too. Such analysis would provide better insights into the sensitivity and specificity of this method, which are critical for determining their clinical utility and comparing these methods. Would the authors explain why they didn’t consider including that analysis. Also, the authors could specify why different threshold determination methods were used for lymph node and tumor samples.

Thank you for your insightful analysis of our work. We did indeed consider a CK19-based analysis, but ultimately decided against it as these types of analyses are prone to false positives when performed on hilar lymph nodes, where parenchymal contamination can occur. However, since the present analysis was performed in mediastinal lymph nodes, we determined that a direct comparison would not be meaningful.

CK19 is positive in the lung parenchyma and bronchial epithelium, whereas NUF2 is positive only in cancerous tissue (as we show in Figure 1). This provides a clear advantage for determining hilar lymph node metastasis, which is necessary to determine indications for sublobar resection. In fact, when we compared survival with CK19 expression in a cohort of patients with lymph node metastasis, we found that it was not actually a significant prognostic factor. These data were not presented because they fell outside the main focus of this study.

Sensitivity and specificity in positive lymph nodes were evaluated at the margins of the dissected lymph nodes, so as not to interfere with the pathological diagnoses performed in this study. Testing for false-negative results was not performed because these were expected to be more frequent. However, we certainly plan to discuss this issue in our future studies, after resolving the associated ethical issues.

The thresholds in this study were divided by the mean value in the tumor area, in order to determine the prognostic value depending on the number of tumors. For lymph node evaluation, ROC curves were used to establish a threshold for predicting recurrence.

  1. Results: Are results presented clearly and analyzed appropriately? Do the conclusions adequately tie together the other elements of the paper?

I commend the authors for their comprehensive and richly informative manuscript, particularly for their extensive use of tables and graphs. It would be more appropriate to maintain a consistent style for all tables. Therefore, I suggest that Table 6 be reorganized accordingly. Additionally, synchronizing the "number at risk" data in the Kaplan-Meier graphs with the timepoints would enhance clarity.

Thank you for this suggestion. We have reorganized the figures and tables accordingly.

In patients with negative lymph nodes, the data on recurrence risk based on NUF2 expression show promising potential for predicting especially lymph node recurrence. However, the sensitivity of this marker in detecting recurrence is approximately 45%, while the specificity for detecting lymph node recurrence is around 70%. Whether this marker alone is sufficient for effective decision-making regarding adjuvant therapy needs to be reassessed from this perspective.

Thank you for your astute observation. In order to minimize interference with the pathological diagnosis, our evaluation was performed at the margins of the dissected lymph nodes. As a result, we considered the associated sensitivity to be low. If the ethical issues surrounding this type of analysis are resolved in the future and evaluation of the entire lymph node becomes possible, the sensitivity will undoubtedly improve.

Analyzing the association between NUF2 expression and tumor size and lymph node positivity provided valuable insights from the study data highlighting the importance of NUF2 expression regardless of these factors. Adding a comparison with stage could further enhance the understanding if these data are available.

Thank you for this insightful proposition. Lung cancer staging is primarily defined by tumor size, pleural invasion, and lymph node metastasis. Therefore, tumor size, pleural invasion, and lymph node metastasis were included in our analysis as substitutes for staging, in order to avoid duplication.

  1. Implications for research, practice and/or society: Does the paper identify clearly any implications for research, practice and/or society? Does the paper bridge the gap between theory and practice? How can the research be used in practice (economic and commercial impact), in teaching, to influence public policy, in research (contributing to the body of knowledge)? What is the impact upon society (influencing public attitudes, affecting quality of life)? Are these implications consistent with the findings and conclusions of the paper?

The implications and phases of the conducted framework provide promising insights. We still need further prospective controlled studies to use this data in clinical practice. It is recommended to provide data on circulating tumor DNA (ctDNA), which is one of the rapidly integrating markers into current clinical practice.

Thank you for this excellent proposal. We will certainly consider performing ctDNA-based analyses in future studies.

  1. Quality of Communication: Does the paper clearly express its case, measured against the technical language of the fields and the expected knowledge of the journal's readership? Has attention been paid to the clarity of expression and readability, such as sentence structure, jargon use, acronyms, etc.

The language used in the article is fluent and understandable, and apart from a few typos, there do not seem to be any major problems.

-Line 145. Figure 1. NUF2 expression in resected lung cancer tissues and adjacent normal lung tissues using quantitative reverse transcription polymerase chain reaction. Bar height indicates average value; whiskers ‘indicatestandard’ error.

Reviewer 3 Report

Comments and Suggestions for Authors

"NUF2 expression in cancer tissues and lymph nodes suggests post-surgery recurrence of non-small cell lung cancer" is an original article about the prognostic value of NUF2 expression in lung cancer tissue and lymph nodes. The paper is well designed and well written, and the results are interesting, as the Authors state that NUF2 expression in negative lymph nodes has prognostic value, and suggest its application to OSNA for the evaluation of sentinel lymph node. I have only few observations:

1) When the Authors state that the application of C19 to OSNA may lead to false positive results (Introduction lines 43-44 and Discussion lines 232-233), a reference is needed.

2) The Authors should report more clearly  the results related to NUF2 expression in lymph nodes. How many cases resulted positive? The Authors suggest that NUF2 could be used for OSNA, so it could be interesting to know the number of NUF2 "false positive" lymph nodes. The Authors should comment the NUF2 "flase positive" lymph nodes in the Discussion, as a possible limitation for OSNA.

Author Response

NUF2 expression in cancer tissues and lymph nodes suggests post-surgery recurrence of non-small cell lung cancer" is an original article about the prognostic value of NUF2 expression in lung cancer tissue and lymph nodes. The paper is well designed and well written, and the results are interesting, as the Authors state that NUF2 expression in negative lymph nodes has prognostic value, and suggest its application to OSNA for the evaluation of sentinel lymph node. I have only few observations:

1) When the Authors state that the application of C19 to OSNA may lead to false positive results (Introduction lines 43-44 and Discussion lines 232-233), a reference is needed.

Thank you for this observation. CK19 is positive in the lung parenchyma and bronchial epithelium, whereas NUF2 is positive only in cancerous tissues (as we show in Figure 1). Therefore, analyses of NUF2 possess clear advantages in terms of determining hilar lymph node metastasis, which in turn is necessary for determining any indications for sublobar resection. However, there are currently no articles in the literature (to the best of our knowledge—and we would certainly be grateful if the reviewer would point us toward one that we are unaware of) that report the false-positive rate when CK19 is used, as no prior studies have performed OSNA analysis on hilar and mediastinal lymph nodes separately. We certainly plan to further examine this issue in our future studies.

2) The Authors should report more clearly  the results related to NUF2 expression in lymph nodes. How many cases resulted positive? The Authors suggest that NUF2 could be used for OSNA, so it could be interesting to know the number of NUF2 "false positive" lymph nodes. The Authors should comment the NUF2 "flase positive" lymph nodes in the Discussion, as a possible limitation for OSNA.

Thank you for your valuable suggestions. Since recurrence was used to determine positivity in this study, we concluded that it would be impossible to determine the sensitivity of this method to the presence of micro-lymph node metastasis. OSNA evaluations using NUF2 should certainly be explored and discussed in future studies, using methods of analysis that include pathologically-confirmed metastasis-positive lymph node specimens.

Round 2

Reviewer 1 Report

Comments and Suggestions for Authors

Thank you for making revisions to your manuscript and resubmitting it. However, I noticed that the revised version did not thoroughly address the previously raised concerns or provide supplementary experimental data as suggested. It is essential that the revised version includes quantitative statistical differences in the immunohistochemistry and fluorescence data. I recommend providing clear evidence of these differences to strengthen the scientific rigor of the study. I recommend that you carefully consider the previous feedback and make necessary revisions and additions to the manuscript based on the review comments.

Author Response

Thank you for your pertinent comments. We performed immunohistochemical analyses on lung cancer tissues, and most of them were positive results. Since quantitative evaluation was difficult, we evaluated the RNA expression levels in this study. We have commented on the lack of immunostaining quantification as a limitation.

Reviewer 2 Report

Comments and Suggestions for Authors

This paper is acceptable now for publication.

Author Response

Thank you for your comments.